# Osteopathic Manipulation of the Sphenopalatine Ganglia Versus Sham Manipulation, in Obstructive Sleep Apnoea Syndrom: A Randomised Controlled Trial

**DOI:** 10.3390/jcm11010099

**Published:** 2021-12-24

**Authors:** Valérie Attali, Olivier Jacq, Karine Martin, Isabelle Arnulf, Thomas Similowski

**Affiliations:** 1UMRS1158 Neurophysiologie Respiratoire Expérimentale et Clinique, INSERM, Sorbonne Université, 75005 Paris, France; jacqolivier.osteo@gmail.com (O.J.); thomas.similowski@upmc.fr (T.S.); 2AP-HP, Département R3S, Service des Pathologies du Sommeil, Hôpital Pitié-Salpêtrière, Sorbonne Université, 75013 Paris, France; isabelle.arnulf@aphp.fr; 3AP-HP, Unité de Recherche Clinique PSL-CFX, Hôpital Pitié Salpêtrière, INSERM, CIC-1901, Sorbonne Université, 75013 Paris, France; karine.martin@aphp.fr; 4Institut du Cerveau-Paris Brain Institute-ICM, INSERM, CNRS, Sorbonne Université, 75013 Paris, France; 5AP-HP, Département R3S, Hôpital Pitié-Salpêtrière, Sorbonne Université, 75013 Paris, France

**Keywords:** obstructive sleep apnea syndrom, sphenopalatine ganglion block, osteopathic physicians, nasal obstruction

## Abstract

(1) Background: osteopathic manipulation of the sphenopalatine ganglia (SPG) blocks the action of postganglionic sensory fibres. This neuromodulation can reduce nasal obstruction and enhance upper airway stability. We investigated the manipulation of the SPG in 31 patients with obstructive sleep apnoea syndrome (OSAS); (2) Methods: Randomised, controlled, double-blind, crossover study. Participants received active (AM), then sham manipulation (SM), or vice versa. The primary endpoint was apnoea-hypopnoea index (AHI). Secondary endpoints were variation of nasal obstruction evaluated by peak nasal inspiratory flow (PNIF) and upper airways stability evaluated by awake critical closing pressure [awake Pcrit]), at 30 min and 24 h. Schirmer’s test and pain were assessed immediately post-manipulation. Tactile/gustatory/olfactory/auditory/nociceptive/visual sensations were recorded. Adverse events were collected throughout. (3) Results: SPG manipulation did not reduce AHI (*p* = 0.670). PNIF increased post-AM but not post-SM at 30 min (AM-SM: 18 [10; 38] L/min, *p* = 0.0001) and 24 h (23 [10; 30] L/min, *p* = 0.001). There was no significant difference on awake Pcrit (AM-SM) at 30 min or 24 h). Sensations were more commonly reported post-AM (100% of patients) than post-SM (37%). Few adverse events and no serious adverse events were reported. (4) Conclusions: SPG manipulation is not supported as a treatment for OSAS but reduced nasal obstruction. This effect remains to be confirmed in a larger sample before using this approach to reduce nasal congestion in CPAP-treated patients or in mild OSAS.

## 1. Introduction

Obstructive sleep apnoea syndrome (OSAS) is a chronic respiratory disease, characterised by repeated intermittent upper airway obstruction during sleep [1], and responsible for cardiovascular, neurococognitive and postural comorbidities [1,2]. An apnoea-hypopnoea index (AHI) > 20.6/h is independently associated with hypertension or metabolic syndrome [3] requiring therapeutic management. Nocturnal continuous positive airway pressure (CPAP) remains the current standard of care, but long-term adherence may be poor [4].

The pathophysiology of OSAS is complex, involving anatomical factors resulting in upper airway narrowing [5], as well as non-anatomical factors of instability such as central control of upper airway muscles [5,6]. Both types of factors may co-exist in the same patient. This finding has led to the recent emergence of personalised medicine [7] and phenotypic approaches to treatment [8,9,10]. In a pilot study, we showed that osteopathic manipulation of the sphenopalatine ganglia (SPG) can enhance upper airway stability in some OSAS patients [11]. In the light of these results, this technique warranted further investigation in the context of a personalised therapeutic approach to OSAS. The SPG are located in the left and right pterygopalatine fossae, posterior to the maxillary sinus. Parasympathetic sensory afferent fibres synapse in these ganglia, which give rise to postganglionic fibres that then travel with the branches of the trigeminal nerve to supply, in particular, the nasal mucosa, nasopharynx and certain upper airway dilator muscles [12]. The presumed mechanism of action of osteopathic manipulation of the SPG, as demonstrated in the management of cluster headache [13], consists of blocking the action of postganglionic sensory fibres. In OSAS, this neuromodulation could reduce the AHI by stabilising the upper airways and by reducing nasopharyngeal congestion. We tested the hypothesis in OSAS patients.

## 2. Materials and Methods

### 2.1. Study Design and Participants

This randomised, controlled, double-blind, crossover study compared single application of active manipulation of the SPG (AM) with sham manipulation (SM). This study was registered on 2 April 2013, under reference NCT01822743, in the clinicaltrial.gov registry. This study was conducted in France in the Department of Sleep Medicine at the Pitié-Salpêtrière Hospital. This study was approved by the *Comité de Protection des Personnes Ile-de-France VI* (Ethics Committee), Paris, France. All methods were performed in accordance with the relevant guidelines and regulations. Participants were recruited between February 2012 and December 2013.

Male and female adult OSAS patients aged > 18 years with an AHI ≥ 15/h and ≤ 45/h, and a body mass index (BMI) ≤ 40 kg/m^2^, were included in this study. Pregnant women or nursing mothers, subjects with complete nasal obstruction, history of pharyngeal surgery, or ongoing treatment with serotonin reuptake inhibitors were not included. All participants received detailed information about the study, informed consent was obtained from all participants, and all provided their written consent to participate in this study.

Of note, the manipulation of the SPG is largely used by osteopaths in non-OSAS patients, for the treatment of temporomandibular disorders [14]. Before studying the OSAS patients, we assessed the effects of the manipulation on nasal patency and upper airways stability in 21 healthy subjects aged 18 to 40 years, having a low probability of OSAS on the Berlin score [15], and with a Body Mass Index < 30 kg/m^2^. The corresponding data are provided in Appendix A.

### 2.2. Study Procedures

The procedure for OSAS patients is on Figure 1. The baseline “pre-manipulation 1” polysomnography was performed in the hospital. Seven days later, the randomisation was performed. Nasal obstruction was then assessed by measuring peak nasal inspiratory flow (PNIF) and upper airway stability was assessed by measuring awake critical closing pressure (awake Pcrit). The first manipulation (active or sham) following randomisation was then administered. Correct application [11] of manipulation was verified by immediately evaluating lacrimation by Schirmer’s test [16] and pain experienced by the patient was assessed by a visual analogue scale. PNIF and awake Pcrit were measured again at 30 min. “Post-manipulation 1” videopolysomnography was performed in the evening. PNIF and awake Pcrit measurements were repeated 24 h after manipulation. The patient was convened 21 days later for “pre-manipulation 2” baseline polysomnography. Seven days after polysomnography, the patient was reviewed for manipulation 2 (SM when the patient had previously received AM of the SPG, or AM when the patient had previously received SM). Manipulation 2 was evaluated in the same way as for manipulation 1. Patients already treated by CPAP or mandibular advancement device (MAD) were asked to stop this treatment for a period of seven days, prior to each polysomnography. In healthy subjects, the same evaluations were performed as for OSAS patients before and 30 min after manipulation, but without the 24-h evaluation for polysomnography, PNIF and Pcrit. A 21-day interval was observed between the two manipulations.

### 2.3. Interventions

AM and SM consisted of purely manual pressure, as described in Jacq et al. [11]. Briefly, AM was performed successively on the right and left SPG, with the subject in the supine position, with the head turned towards the opposite side to that of the SPG to be manipulated. The osteopath held the subject’s head with one hand and introduced the little finger of the other hand into the subject’s mouth directed towards the pterygoid process to reach the pterygopalatine fossa to apply pressure to the SPG for several seconds (Figure 2).

SM consisted of introducing the little finger into the subject’s mouth, in the same direction as for active manipulation, but without reaching the pterygoid process, and applying pressure to the mandible.

### 2.4. Outcomes

#### 2.4.1. Primary Endpoint

AHI obtained by videopolysomnography approximately six hours after AM or SM. All polysomnographies were performed in hospital with recording of electroencephalography (three leads: Fp1-Cz, C3-A2, and O1-Cz), left and right electrooculography, surface mentalis electromyography, electrocardiography, nasal pressure, assessment of thoracic and abdominal efforts by plethysmographic belts and transcutaneous oximetry. Respiratory events were interpreted according to the American Academy of Sleep Medicine manual for the scoring of sleep and associated events [17].

#### 2.4.2. Secondary Endpoints

PNIF was measured by a peak flow meter (In-Check inspiratory flow meter; AllianceTech Medical, Granbury, TX, USA) using a face mask covering the nose and mouth, applied tightly to the face. The participant was asked to inhale deeply and rapidly. This was repeated three times and the higher value measurement was recorded. The PNIF is a reproducible method [18] and normative values are available [19]. The PNIF is correlated to acoustic rhinometry [20] and is more sensitive than rhinomanometry to “decongesting” manoeuvres [21]. The PNIF then represents an adequate method to assess the potential decongesting effect of the osteopathic manipulation.

Awake Pcrit was defined as the negative pressure required to induce upper airway obstruction and zero air flow. It was estimated experimentally according to a procedure identical to that described in the study by Jacq et al. evaluating active SPG manipulation in OSAS patients [11]. Awake Pcrit has been validated for the evaluation of upper airway collapsibility in healthy subjects [22] and to assess the effect of mandibular advancement device in OSAS patients [23]. In our previous pilot study, we showed that the manipulation of the SPG improved significantly upper airway stability as assessed by awake Pcrit [11].

Tear production was assessed by Schirmer’s test [16], as described in Jacq et al. [11]. Pain assessed by a pain visual analogue scale (in mm) and sensory perception assessed by a questionnaire (*Did you experience any gustatory, olfactory, visual, auditory, tactile, or nociceptive sensations?*), were administered immediately after manipulation. Total sleep time (TST), % of N1, N2, N3, REM sleep stages, desaturation index, time spent at SpO_2_ <90%, were analysed.

### 2.5. Randomisation and Masking

The order of administration of AM and SM (AM in period 1 and SM in period 2, or SM in period 1 and AM in period 2) was determined by randomisation. All participants therefore received AM and SM. Two investigators participated in the study procedures, an osteopath and a physician. Randomised allocation was centralised through a web server and only the investigating osteopath performing manipulation was informed of the result. The osteopath was not involved in either clinical evaluation or data analysis. The investigating physician was not informed about the results of randomisation or whether the participant received AM or SM. The physician did not administer any manipulation and conducted clinical evaluations in a blinded manner.

### 2.6. Statistical Analysis

The primary endpoint was the (SM–AM) difference of AHI variations (%), between baseline and post-manipulation polysomnography. The expected reduction in AHI was 25% after AM and 5% after SM. The estimated standard deviation of the difference was 30%. For a type I error of 5%, 26 patients would ensure a power of 80% of the study. As the patient dropout rate was estimated to be a maximum of 10%, a total of 30 patients had to be included in the study.

All patients were included in the tolerance population. Participants who received both AM and SM were included in the main analysis. As not all variables had a normal distribution, data were expressed as median and interquartile range. Continuous variables were compared by Wilcoxon rank tests for paired data and proportions were compared by Fisher’s exact test. All tests were two-tailed, and *p*-values < 0.05 were considered statistically significant. The carry-over and period effects were tested. Statistical analyses were performed using SAS software, Version x.y (SAS Institute, Cary, NC, USA). Of note, the population of healthy subjects was included for descriptive purposes. As no prior data were available in this population, the sample size was not calculated on the basis of the effect of AM and we estimated that the inclusion of 20 healthy subjects would be sufficient to obtain descriptive data. Furthermore, no comparison was planned between OSAS patients and healthy subjects.

## 3. Results

In this case, 31 OSAS patients were included. One patient was randomised, but then immediately withdrew his consent and did not receive either of the two manipulations. In this case, 30 patients completed the study and were analysed for all criteria (Figure 3). In this case, 15 of these 30 patients had been previously treated by CPAP (*n* = 12) or MAD (*n* = 3). Nine of these 15 patients were able to discontinue their treatment in line with the study requirements and six patients presented a protocol violation in relation to this criterion (no discontinuation of treatment during the study (*n* = 1), satisfactory discontinuation of treatment for AM, but not for SM (*n* = 4), satisfactory discontinuation of treatment for SM, but not for SM (*n* = 1)). The baseline characteristics are presented in Table 1. Twenty-one healthy subjects were included. One healthy subject was randomised and re-ceived AM, but was subsequently lost to follow-up and did not receive SM (see consort flow diagram on Appendix A). Baseline characteristics are in Appendix A.

### 3.1. Primary Endpoint

The observed difference with respect to baseline AHI was −2.3% (−24.9; 22.6) after AM and 0.9% (−27.8; 21.1) after SM. AM did not induce any reduction of the AHI compared to SM (*p* = 0.670) (Figure 4). The night to night variability of AHI was of 19% [10; 35]. There was no evidence of a carryover effect (*p* = 0.35) or a period effect (*p* = 0.57).

### 3.2. Secondary Endpoints

PNIF increased by 15 [0; 24] L/min, 30 min after AM and decreased by 5 [−20; 0] L/min after SM (AM-SM 18 [10; 38] L/min; *p* = 0.0001). Similar results were observed at 24 h: after AM: +23 [10; 30] L/min, after sham SM: 0 [−10; 10] L/min, (AM-SM 25 [10; 41] L/min; *p* = 0.0001) (Figure 5).

Awake Pcrit was uninterpretable in nine patients due to lack of cooperation during the measurement. Analysis was therefore performed on a subgroup of 21 OSAS patients The AM-SM difference was −2.3 [−16.4; 8.4] cm H_2_O (*p* = 0.95) at 30 min and −2.2 [−16.3; 32.9] cm H_2_O at 24 h (*p* = 0.71). No difference was observed between AM and SM in terms of total sleep time, percentage of the various sleep stages, desaturation index, and time spent at SpO_2_ < 90% (Table 2).

AM induced a significant increase in tear production compared to SM (difference [25th–75th percentiles]: 0.12 mm/s [0.07; 0.23]; *p* = 0.001) Pain was significantly more intense after AM than after SM (AM-SM difference [25th–75th percentiles]: 76% [63; 89]; *p* < 0.0001 AM induced at least one sensation in 100% of patients, while SM induced a sensation in only 11 patients (37%) (AM versus SM, *p* < 0.0001). A significantly higher number of sensations per subject was reported after AM than after SM (3 [2; 3] versus 0 [0; 1] *p* < 0.0001). The sensations reported after AM were (% of patients), nociceptive (87%), tactile (77%), olfactory (50%), gustatory (27%), auditory (20%), visual (13%). Very few sensations were reported after SM in the two populations and mainly consisted of tactile (sensation of the latex glove on the mandible), or olfactory (latex smell) sensations. One patient reported mild pain after SM. Analysis of free verbatim descriptions showed that 8 patients reported a taste of blood in the absence of any wounds in the mouth. In this case, 15 patients reported changes in the perception of the position of the mandible or temporomandibular joints after AM. Five patients described ear sensations (blocked and/or unblocked ears, ringing in the ears, transient tinnitus) after AM. In this case, 12 patients described improved nose breathing, while one patient reported nasal obstruction after AM. Three patients (10%) described pharyngeal sensations after AM.

In healthy subjects, no difference between AM and SM was observed on PNIF at thirty minutes (AM-SM 10 L/min [−25; 15]; *p* = 0.6614). Awake Pcrit was analysable in 17 healthy subjects. The AM-SM difference was 0.3 [−7.1; 6.3] cm H_2_O at thirty minutes (*p* = 0.85). AM induced a significant increase in tear production compared to SM (difference [25th–75th percentiles]: 0.23 mm/s [0.09; 0.40]; *p* = 0.001). Pain was significantly more intense after AM than after SM (AM-SM difference [25th–75th percentiles]: 67% [62; 77]; *p* < 0.0001). AM induced at least one sensation in 100% of healthy subjects, while SM induced a sensation in only 6 subjects (29%) (AM versus SM, *p* < 0.0001). A significantly higher number of sensations per subject was reported after AM than after SM (3 [2; 3] versus 0 [0; 1]; *p* < 0.0001). The sensations reported after AM were (% of subjects), nociceptive (62%), tactile (86%), olfactory (24%), gustatory (62%), auditory (14%), visual (10%). Very few sensations were reported after SM and mainly consisted of tactile (sensation of the latex glove on the mandible), or olfactory (latex smell) sensations. Analysis of free verbatim descriptions showed that 12 healthy subjects reported a taste of blood in the absence of any wounds in the mouth. Sixteen healthy subjects reported changes in the perception of the position of the mandible or temporomandibular joints after AM. Three healthy subjects described ear sensations (blocked and/or unblocked ears, ringing in the ears, transient tinnitus) after AM. Two healthy subjects described improved nose breathing, while one subject reported nasal obstruction after AM.

### 3.3. Adverse Effects

No serious adverse events were reported during the study. Four patients (13%) reported mild jaw pain or discomfort 24 h after AM. No adverse effects were reported after SM.

## 4. Discussion

This study shows that single active osteopathic manipulation of the SPG does not reduce the AHI in patients with moderate-to-severe OSAS despite an increase in nasal airflow in these patients.

### 4.1. Effects of AM on the Upper Airways and OSAS

PNIF values measured in OSAS patients were within the normal range [24,25]. The statistically significant increase in nasal airflow following AM suggests the probable presence of mild nasal obstruction in these patients at baseline [24]. The reversible nature of this nasal obstruction suggests a congestive origin [26], consistent with the previously reported association between nasal congestion and sleep-disordered breathing [27]. In terms of pathophysiology, nasal obstruction predisposes to pharyngeal instability during sleep, as the upper airways can be likened to a Starling model comprising a flexible midline pharyngeal structure surrounded by two rigid structures, the nasopharynx proximally and the trachea distally [28]. In this context, improvement of nasal airflow could constitute a theoretical target in the treatment of OSAS [29]. However, nasal congestion appears to play a minor role in the pathophysiology of OSAS, probably due to the heterogeneity of the disease and the multiple anatomical and non-anatomical factors that may contribute to upper airway instability during sleep [5]. In our study, although AM of the SPG resulted in a reduction of nasal congestion comparable to that obtained by drug treatments for rhinosinusitis [24], or nasal polyposis [30], this effect was not associated with correction of the AHI. This result is consistent with the results reported by the majority of studies targeting nasal obstruction in the treatment of OSAS [26,31,32]. The improvement of nasal flow that we describe does have some relevance. The treatment of nasal obstruction is a major goal in the management of OSAS. It is recognized that reducing the nasal obstruction is not a specific treatment of OSAS but improve quality of life and snoring [33]. Moreover, it helps reduce the level of CPAP necessary to control the AHI and it improves CPAP tolerance and use [34]. Nasal obstruction is also known to negatively impact the outcome of treatment with mandibular advancement devices [35].

In addition, we did not observe any increase in upper airway stability measured by awake Pcrit after AM, which is not surprising in healthy subjects, but more surprising in OSAS patients in the light of our previously reported results [11]. This variable must be interpreted very cautiously, as awake Pcrit could not be measured in about one-third of OSAS patients due to lack of cooperation and marked intra-individual variability in our population. However, our results are not in favour of an effect of AM on reduction of pharyngeal stability.

### 4.2. Mechanism of Action of AM

SPG neuromodulation by pharmacological blockade, surgical ablation, or implanted stimulation is proposed for the management of refractory head and neck pain, including cluster headache, trigeminal neuralgia, or post-sinus surgery headache [13]. The mechanism of action in these diseases is blockade of the trigeminovascular network of which the SPG is a component and, more precisely, blockade of the parasympathetic autonomic vasomotor system that is responsible, among other things, for nasal congestion [12]. In contrast, in stroke, SPG neuromodulation by implanted stimulation targets a stimulant effect on the parasympathetic nervous system [36] with a demonstrated benefit in terms of increased cerebral blood flow secondary to the release of vasodilator neurotransmitters (Nitric Oxide, acetylcholine), decreased cerebral oedema due to stabilisation of the blood-brain barrier, and potentially a neuroprotective effect [37]. In particular, it should be noted that high-frequency implanted SPG stimulation induces a reduction of postganglionic parasympathetic activity [13], while a parasympathetic stimulant effect is observed in response to low-frequency stimulation, with the consequent risk of cluster-like attacks [38]. Although the mechanism of action of AM cannot be confirmed in our patients, the induced reduction in nasal congestion may indicate neuromodulation secondary to postganglionic parasympathetic blockade. This hypothesis is supported by the associated neurosensory manifestations reported after AM, some of which are similar to those reported during parasympathetic blockade by implanted stimulation in cluster headache [39].

### 4.3. Methodological Considerations

A strength of this study is the randomised, controlled, double-blind methodology [40]. In contrast with studies testing drug treatments, use of manual therapy did not allow blinding of the osteopath, but this difficulty was circumvented by separating the interventions between an osteopath applying the stimulation, but not performing any evaluation, and a physician blinded to the randomisation arm, who ensured blind evaluation of the outcome measures. We acknowledge that the study was not triple-blinded (osteopath, physician, patient), In this context, the study can be considered to have been conducted under double-blind conditions from the point of view of the operator/investigator pair. The whole procedure, including the choice of sham manipulation (same position, same instructions, same duration), its ability to ensure maintenance of blind conditions, and the criteria ensuring correct administration-induction of lacrimation and brief pain- had been previously validated in a pilot study [11]. In this pilot study, the efficacy of the blinding was assessed with the answer to the following question at the end of the study “*Which manipulation was the active one?*”. In this case, 33% of patients were not able to determine which manipulation (active or sham) was the active one, and the remaining patients were all wrong. We are confident that the absence of effect of the manipulation of AHI was not biased. The night-to-night variability of the AHI was as expected around 20%. The cross-over design with a 3-week wash-out period was validated in our previous pilot study [11], and the present trial confirmed the absence of carry-over and period effects. This study also presents several limitations, including the impossibility of obtaining Pcrit data in one-third of patients, as already mentioned. Furthermore, the study does not provide any mechanistic arguments concerning the mode of action of AM, and the hypothesis of a neuromodulatory action remains putative, as we did not measure parasympathetic activity directly. The intra-individual variability of the response to the manipulation was not assessed in this study and remains to be studied particularly on nasal obstruction. Finally, we only studied a single SPG manipulation and cannot exclude the possibility that different results could have been obtained after repeated manipulation.

## 5. Conclusions

This is one of the rare studies to apply state-of-art clinical trial methodology to an osteopathic approach. It shows that SPG manipulation does not reduce the AHI in OSAS patient. Nevertheless, it shows that SPG manipulation reduced nasal congestion in a small group of patients with moderate-to-severe OSAS. Reducing nasal congestion is a relevant issue in CPAP-treated OSAS patients [41] or as an adjunctive measure in patients with mild OSAS, however our results need to be confirmed in a larger population of OSAS patients, before using this osteopathic approach a routine practice.

## Figures and Tables

**Figure 1 jcm-11-00099-f001:**
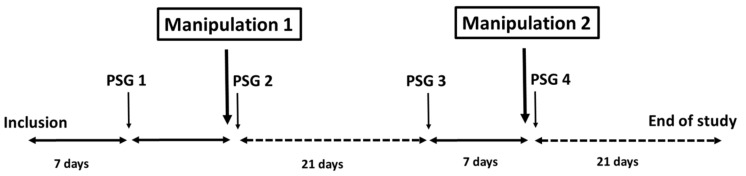
Study plan for OSAS patients. PSG: polysomnography.

**Figure 2 jcm-11-00099-f002:**
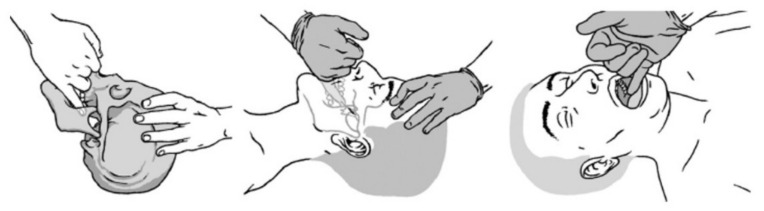
Active manual stimulation of the sphenopalatine ganglion. Reproduced from Jacq et al. BMC Complementary and Alternative Medicine (2017) 17:546; DOI 10.1186/s12906-017-2053-0.

**Figure 3 jcm-11-00099-f003:**
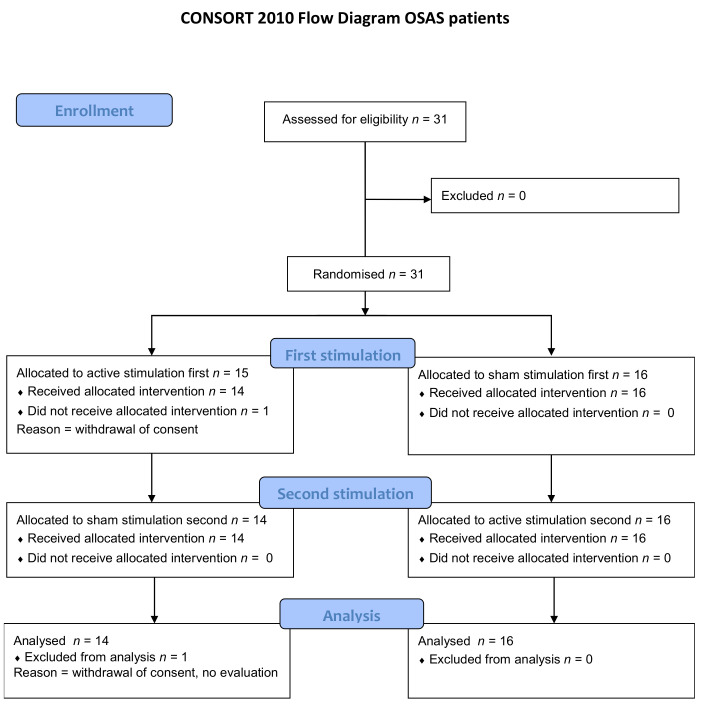
Consort Flow diagram for OSAS patients.

**Figure 4 jcm-11-00099-f004:**
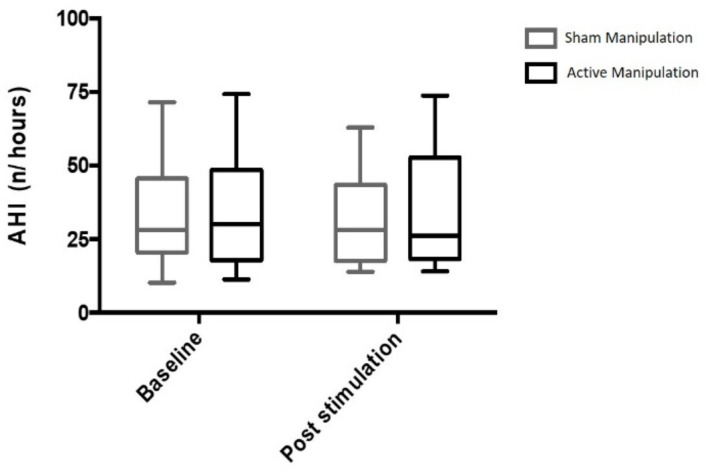
Variation of the apnoea-hypopnoea index after active manual stimulation of the sphenopalatine ganglion and after sham manual stimulation in OSAS patients. Results are median and 10–90 centiles.

**Figure 5 jcm-11-00099-f005:**
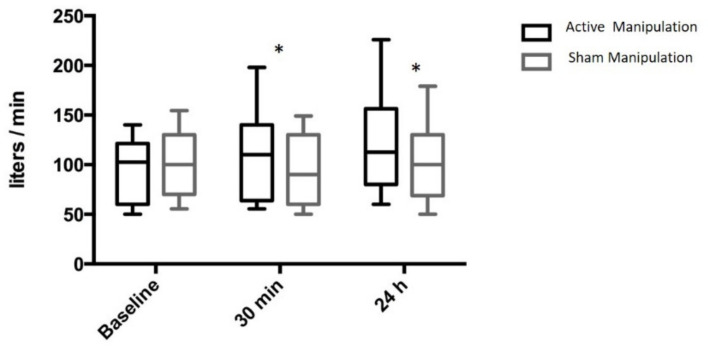
Variation of peak nasal inspiratory flow after active manual stimulation of the sphenopalatine ganglion and after sham manual stimulation in OSAS patients. Results are median and 10–90 centiles. * *p* = 0.0001 Active versus Sham manipulation.

**Table 1 jcm-11-00099-t001:** Baseline data before active manual stimulation of the sphenopalatine ganglion and before sham manual stimulation.

	OSAS Patients (*n* = 30)	
Age (Years)	57 [33; 64]	
Gender (Male/Female; *n*)	24/6	
	Before AM	Before SM
Weight (kg)	89 [78; 103]	90 [78; 102]
BMI (kg/m^2^)	28.3 [27.1; 34,4]	29.1 [27; 35]
ESS (/24)	7 [4; 13]	9 [5; 12]
PNIF (L/min)	103 [60; 120]	100 [70; 130]
Awake Pcrit	−19.6 [−31.7; −12.9]	−25.6 [−39.2; −16.5]
Polysomnography		
TST (min)	443 [388; 489]	452 [391; 499]
Sleep efficiency (%)	88 [79; 93]	85 [78; 91]
LSO (min)	24 [14; 40]	16 [9; 25]
Arousal Index (n/h)	26 [15; 36]	26 [19; 40]
N1 sleep (min)	6 [4; 10]	5 [3; 11]
N2 sleep (min)	250 [219; 289]	251 [187; 300]
N3 sleep (min)	81 [58; 108]	70 [53; 102]
REM sleep (min)	98 [78; 117]	100 [77; 139]
AHI (n/h)	30 [18; 48]	28 [21; 44]
AHI ≥ 30 (% patients)	50	47
SpO_2_ < 90% (%TST)	7 [1; 15]	4 [1; 14]

AM, Active Manipulation; SM, Sham Manipulation; BMI, Body Mass Index; ESS, Epworth Sleepiness Scale; PNIF, Peak Nasal Inspiratory Flow; TST, Total Sleep Time; LSO, Latency to sleep onset; REM Sleep, Rapid Eyes Movement Sleep Stage; AHI, Apnoea-hypopnoea Index.

**Table 2 jcm-11-00099-t002:** Variation of polysomnographic data after manipulation in OSAS patients.

	After AM	After SM	*p*
Total Sleep Time (min)	19 [−25; 59]	12 [−31; 52]	0.68
N1 sleep (min)	0 [−3; 6]	−1 [−4; 4]	0.57
N2 sleep (min)	−6 [−23; 40]	−10 [−35; 18]	0.21
N3 sleep (min)	1 [−17; 20]	10 [−24; 42]	0.42
REM sleep (min)	14 [−3; 23]	13 [−8; 46]	0.54
Arousal Index (n/h)	2 [−4; 5]	−3 [−7; 4]	0.37
Apnea-Hypopnea Index (%)	−2.3 [−24.9; 22.6]	0.9 [−27.8; 21.1]	0.67
Desaturation Index (%)	0.0 [−19.6; 38.7]	0.0 [−13.5; 21.0]	0.94

AM, Active Manipulation; SM, Sham Manipulation; REM Sleep, Rapid Eyes Movement Sleep Stage.

## Data Availability

The study protocol is available at: https://www.researchgate.net/project/Manual-Stimulation-of-Sphenopalatine-Ganglia-and-Obstructive-Sleep-Apnea-Syndrom (accessed on 20 December 2021). Individual participant data that underlie the results reported in this article, after de-identification (text, tables, figures, and appendices). Proposals should be directed to valerie.attali@aphp.fr to gain access, data requestors will need to sign a data access agreement.

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
