# Peer review of "Osteopathic Manipulation of the Sphenopalatine Ganglia Versus Sham Manipulation, in Obstructive Sleep Apnoea Syndrom: A Randomised Controlled Trial"

_jcm, 2021, doi:10.3390/jcm11010099_

Round 1
Reviewer 1 Report
The present study assessed the effect of osteopathic manipulation on obstructive sleep apnea. Evaluating osteopathic manipulation is original and of interest. Interventions seem to be well standardized. However, I have several concerns, mainly regarding the study design:
- The main limitation of this paper is the study design chosen. First, it is not clear why the authors chose to include both healthy and OSA patients in the same study while their main outcome is to assess the effect of osteopathic manipulation in OSA patients. By reading the study protocol, I assume that healthy participants were included to assess the effect of manipulation on the stability of the UA. That is a quite different outcome in a different population which deserve a separate study. In my opinion, it would be clearer if only OSA patients are included in this paper.
- Second, why the authors choose a randomised controlled crossover design and not a randomised controlled design? If one assumes that osteopathic manipulation is effective to reduce OSA severity, we may hypothesize that this effect lasts beyond 3 weeks and may interfere with the sham condition if the osteopathic condition is achieved first. What is the opinion of the authors on that point?
- Do you think that the absence of a significant effect of osteopathic manipulation may be due to a too short period between AM and PSG? Is there a minimum time needed for osteopathy to be effective?
- Figure 1 is important but the bottom box is not very clear for me. It took me a while to understand that there was a psg before and after manipulation.
- Although more difficult to assess, sleep Pcrit would better reflect OSA improvement than daytime Pcrit. Do the authors know how daytime Pcrit correlates with sleep Pcrit?
- Table 2. Please include other PSG parameters such as apnea, hypopnea and oxygen parameters.
- I do not agree that this is a double blinded study since the operator knowns in which group is the participant
- conclusion: the authors should nuance their words since they only assess the effect of osteopathic manipulation on one night and the middle and long term effect remain unknwon.
Minot comment:
- statistical analysis methods: p-value threshold is not defined.
Reviewer 2 Report
The approach that is made is novel and of interest.
I consider that the measurement methods are unreliable and the variables are subject to a great dispersion in their values.
Manual intervention is also subject to great variability in the way it is performed as well as the anatomy of each subject.
The sample is small to draw robust conclusions.
In general I consider that negative conclusions are not relevant. The conclusions that are raised regarding the improvement of nasal flow congestion are also not very relevant.
I believe that active anterior rhinomanometry is the best method to assess nasal resistance and that the variability of the nasal cycle of each subject must be taken into account.
I think the statistical method should be specified more
Round 2
Reviewer 2 Report
The main suggestions and considerations have been taken into account.
Some of the objections have been explained and clarified. I still do not agree with some of them but it is not relevant
I think the work has improved significantly
Thank you